# Cortical and Striatal Astrocytes of Neonatal Rats Display Distinct Molecular and Pharmacological Characteristics of Dopamine Uptake

**DOI:** 10.3390/ijms25020911

**Published:** 2024-01-11

**Authors:** Vesna Sočan, Klemen Dolinar, Mojca Kržan

**Affiliations:** 1Institute of Pharmacology and Experimental Toxicology, Faculty of Medicine, University of Ljubljana, 1000 Ljubljana, Slovenia; vesna.socan@mf.uni-lj.si; 2Institute of Pathophysiology, Faculty of Medicine, University of Ljubljana, 1000 Ljubljana, Slovenia; klemen.dolinar@mf.uni-lj.si

**Keywords:** astrocyte, dopamine uptake, NET, PMAT, neonatal rat

## Abstract

Astrocytes are crucial in the regulation of neurotransmitter homeostasis, and while their involvement in the dopamine (DA) tripartite synapse is acknowledged, it necessitates a more comprehensive investigation. In the present study, experiments were conducted on primary astrocyte cultures from the striatum and cortex of neonatal rats. The pharmacological intricacies of DA uptake, including dependence on time, temperature, and concentration, were investigated using radiolabelled [^3^H]-DA. The mRNA expression of transporters DAT, NET, PMAT, and OCTs was evaluated by qPCR. Notably, astrocytes from both brain regions exhibited prominent mRNA expression of NET and PMAT, with comparatively lower expression of DAT and OCTs. The inhibition of DA uptake by the DAT inhibitor, GBR12909, and NET inhibitors, desipramine and nortriptyline, impeded DA uptake in striatal astrocytes more than in cortical astrocytes. The mRNA expression of NET and PMAT was significantly upregulated in cortical astrocytes in response to the DA receptor agonist apomorphine, while only the mRNA expression of NET exhibited changes in striatal astrocytes. Haloperidol, a DA receptor antagonist, and L-DOPA, a DA precursor, did not induce significant alterations in transporter mRNA expression. These findings underscore the intricate and region-specific mechanisms governing DA uptake in astrocytes, emphasizing the need for continued exploration to unravel the nuanced dynamics of astrocytic involvement in the DA tripartite synapse.

## 1. Introduction

Astrocytes, named after their star-like shape, are one of the most abundant cell types in the central nervous system (CNS) [1]. They are crucial regulators of numerous homeostatic functions as they provide structural and metabolic support to neuronal cells, control ion balance, and maintain the blood–brain barrier [2,3]. Unlike neuronal cells, astrocytes cannot generate an action potential but respond to stimuli through fluctuations in intracellular Ca^2+^ [4]. Astrocytes help maintain neurotransmitter homeostasis in the CNS and modulate synaptic transmission. Together with neuronal cells, they form the tripartite synapse, a concept coined by Perea et. al. describing bidirectional communication between astrocytes and neuronal cells [5]. Astrocytes express various transporter proteins in their plasma membrane for neurotransmitter uptake and are also capable of (glio)transmitter release themselves [4,6,7,8,9,10,11,12].

The role of astrocytes in the homeostasis of the neuromodulator dopamine (DA) has been studied to a limited extent. DA is involved in the regulation of several aspects of brain function through multiple DA pathways, each related to a particular brain region [13,14]. DA function in the cortex, particularly the prefrontal cortex, is involved in higher-order cognitive functions, such as attention, working memory, decision-making, and executive control. The mesocortical DA pathway, originating from the ventral tegmental area in the midbrain, projects to the prefrontal cortex, and imbalances in this pathway have been linked to disorders, such as schizophrenia and attention deficit hyperactivity disorder [15]. In contrast, the striatum, a key component of the basal ganglia, is involved in motor control, reward processing, and habit formation. It receives DA input from the nigrostriatal pathway, originating in the substantia nigra. Dysfunctions in the nigrostriatal pathway have been associated with addiction, depression, and other conditions, such as Parkinson’s disease (PD) [13]. Astrocytes have been implicated in the development of and in the protection from these diseases [16,17,18]. Although they have traditionally been viewed as simple, homogenous cells providing support to neurons, we now recognize that astrocytes from different brain regions are heterogeneous [19]. Chai et al. demonstrated that cortical astrocytes and hippocampal astrocytes were similar when comparing RNA sequencing data, but hippocampal and striatal astrocytes were shown to be different populations [20]. Whether astrocytes from distinct brain regions serve different roles in DA homeostasis is yet to be discovered.

DA exerts its effects by binding to metabotropic G protein-coupled DA receptors, which are divided into two subtypes, D1 and D2 receptors [13,21,22]. The DA concentration in the synapse is controlled mainly by its reuptake by the sodium-dependent, low-capacity, high-affinity (uptake 1) dopamine transporter (DAT) [23] and, to a lesser degree, by other transporters. Studies on rodents have reported that the norepinephrine transporter (NET) has an equal affinity for DA and plays a larger role in DA uptake in brain regions where DAT is sparsely distributed, particularly in the cortex [24,25,26] as well as in the DA-denervated striatum in PD [27]. Sodium-independent transporters with a low affinity for DA, albeit high capacity (uptake 2), such as organic cation transporter 3 (OCT3) [28,29] and plasma membrane monoamine transporter (PMAT) [30,31], have an important role in the removal of excess extracellular DA when the DA concentration is high and has overcome the capacity of DAT [27]. They transport organic cations, zwitterions, and some uncharged compounds and operate as facilitated diffusion systems and/or antiporters [32]. OCTs and PMAT are considered “polyspecific” or “multispecific”, as they interact with a wide array of cationic compounds with diverse chemical structures. Nevertheless, there are notable differences in substrate specificity and transport kinetics among these transporters. PMAT has been shown to have a strong kinetic preference for serotonin and DA over other monoamines (i.e., histamine, norepinephrine, and epinephrine), whereas a seemingly opposite preference was observed for OCT3 [33].

Several studies have shown that astrocytes are able to take up DA [21,24,34,35,36,37,38] and the DA precursor, L-DOPA [39,40,41,42,43,44], and express DA receptors in their membrane [35,45,46,47]. Astrocytes respond to DA with fluctuations in intracellular calcium [48,49] and may be involved in response to treatment with dopaminergic drugs, such as DA receptor antagonists, haloperidol and clozapine, as well as the DA receptor agonist, apomorphine [50,51,52,53,54,55,56]. Whether astrocytes are capable of active uptake of DA as well as their role in the homeostasis of DA is up for debate. The exitance of the primary mechanism responsible for DA reuptake into neuronal cells, DAT, has not been found in rat cortical astrocytes [24]; however, it has been found in striatal rat astrocytes [39] and striatal mouse astrocytes [57]. Inazu et al. have shown that DA is transported into rat cortical astrocytes via NET, rather than DAT, as well as by high-capacity, low-affinity uptake 2 transport mediated by OCT3. Contrarily, Hösli et al. showed astrocytic DA uptake in striatal and cerebellar primary cell cultures of neonatal rats is mediated only by facilitated diffusion, and uptake is independent of Na^+^/Cl^−^ ions [34]. As astrocyte morphology has been shown to be region-dependent, particularly in the human brain [58,59], it may be one of the reasons contributing to the conflicting results of astrocyte DA uptake studies.

The aim of our study was to determine the regional pharmacological and molecular characteristics of DA uptake into neonatal rat astrocytes from two distinct brain regions, the cortex and striatum, to further elucidate the role of astrocytes in the homeostasis of DA. Furthermore, we examined whether the drugs apomorphine, L-DOPA, and haloperidol, used in the treatment of pathologies stemming from aberrations in the dopaminergic system, such as PD and schizophrenia, affect the gene expression of these transporters in astrocytes.

## 2. Results

### 2.1. Dependence of [^3^H]-Dopamine Uptake on Time, Temperature, and Concentration in Neonatal Rat Astrocytes from the Cortex and Striatum

We first examined the time course of [^3^H]-DA uptake with a 30 nM concentration of DA in cultured cortical and striatal rat astrocytes for the time span of 60 min. As shown in Figure 1a,b, [^3^H]-DA uptake increased in a time-dependent manner for the initial 20 min and almost reached a plateau at 30 min in both cortical and striatal astrocytes. The accumulation of [^3^H]-DA in both cortical and striatal astrocytes was significantly greater at 37 °C than the accumulation of [^3^H]-DA uptake at 4 °C. Based on these findings, we carried out the following [^3^H]-DA uptake assays for 20 min.

Astrocytes were then exposed to different concentrations of [^3^H]-DA (0.03–1000 µM) for a time span of 20 min, in line with the time course experiments, and the total (37 °C) and the nonspecific (4 °C) DA uptake were measured from which the specific uptake was calculated (as the difference between total and nonspecific uptake) (Figure 1c,d). DA uptake into cultured cortical astrocytes, presented in Figure 1c, was clearly dependent on the DA concentration. However, it does not appear saturable at even the highest concentration of 1 mM, whereas DA uptake in the striatum (Figure 1d) was saturable at 1 mM concentration. The apparent maximum saturation of the radioligand, B_max_, of the specific uptake was determined as 2343 ± 251 pmol/mg.

The DA uptake velocity in striatal astrocytes (Figure 1f) was calculated from the specific uptake and the time span of the [^3^H]-DA incubation (20 min). The kinetic parameters of the uptake velocity were calculated using the Michaelis–Menten equation. The apparent maximal uptake rate, V_max_, of DA was calculated as 117 ± 13 pmol/mg/min and the apparent Michaelis–Menten constant, K_m_, was calculated as 987 ± 176 µM. Cortical astrocytes appear to have a greater uptake capacity for DA in comparison to striatal astrocytes.

### 2.2. Dependence of [^3^H]-Dopamine Uptake in Cortical and Striatal Astrocytes of Neonatal Rats on the Presence of Ouabain amd Sodium Ions 

A 1 mM concentration of the Na^+^/K^+^-ATPase pump inhibitor ouabain reduced 30 nM [^3^H]-DA uptake to 68 ± 2% in cortical and 66 ± 2% in striatal astrocytes in comparison to the control (Figure 2a). In contrast, 30 nM [^3^H]-DA uptake was reduced to 57 ± 8% of the control in cortical astrocytes and to 41 ± 6% of the control in striatal astrocytes in the absence of Na^+^ (Figure 2b). The DA uptake was significantly reduced by both ouabain and the absence of Na^+^. However, there was no significant difference in the DA uptake between the studied brain regions (Student *t*-test, *p* > 0.05).

### 2.3. qPCR Analysis of Transporter mRNA Expression in Neonatal Rat Astrocytes from the Cortex and Striatum

We examined the mRNA expression of high-affinity and low-affinity transporters involved in DA uptake in neonatal rat astrocytes from the cortex and striatum as well as neonatal rat cerebral cortical and striatal tissue. qPCR was performed after the extraction of total RNA from three-weeks-old, confluent cultures and tissue samples for the comparison of transporter mRNA expression between tissue (additionally used as a positive control) and astrocyte cell culture samples.

The mRNA expression profile of the transporters in neonatal rat tissue samples (Figure 3, Tissue) was in line with our expectations. OCT1 and OCT2 mRNA expression was lower than OCT3 mRNA expression in the two studied brain regions. PMAT tissue expression was prominent in both tissue samples. Active uptake transporters, DAT and NET, were expressed in both tissue samples. Conversely, astrocyte cultures from both brain regions displayed a distinct transporter mRNA expression profile, with OCT1 and OCT2 being more prominently expressed than OCT3. Among the studied transporters, PMAT mRNA expression appeared to be the greatest in both brain tissue samples, as well in cortical and striatal astrocytes. The striatal astrocyte mRNA expression of PMAT was statistically significantly greater than the cortical PMAT mRNA expression (*p* < 0.05) (Figure 3, Astrocytes). The mRNA expression of uptake 1 transporter, NET, exceeded the expression of DAT in astrocyte cell cultures from both brain regions, the cortex and striatum. Interestingly, DAT mRNA expression was lower in striatal than in cortical neonatal rat tissue, whereas striatal astrocytes displayed low, but present, mRNA expression of DAT. This was contrary to cortical astrocytes, where DAT mRNA expression was at the limit of detection of our qPCR method (Figure 3, Astrocytes). The differences in transporter expression in astrocytes from each brain region were determined separately by the Brown–Forsythe ANOVA test with post-hoc Dunnett’s T3 multiple comparisons test (Cortex: F = 17.54 (5.00, 35.28), *p* < 0.0001, Striatum F = 23.21 (5.00, 17.05), *p* < 0.0001). Striatal PMAT mRNA expression was significantly greater than all other transporters (*p* < 0.05), whereas NET mRNA expression was significantly greater than DAT, OCT2, and OCT3 (*p* < 0.05). Cortical astrocytes displayed no significant difference between mRNA expression of PMAT and NET; however, the mRNA expression of both transporters was significantly greater than the expression of OCT1, OCT2, OCT3, and DAT.

### 2.4. Inhibition of [^3^H]-Dopamine Uptake by Antidepressants, Desipramine, Nortriptyline, Amitriptyline, DAT Inhibitor GBR12909, Corticosterone, and Decynium 22 

We evaluated the effect of DAT inhibitor GBR12909, OCT and PMAT inhibitors, D22 and corticosterone, and NET selective inhibitors, tricyclic antidepressants, amitriptyline, nortriptyline, and desipramine on DA uptake into striatal and cortical astrocytes. Inhibition curves are presented in Figure 4. The apparent affinity for the transporters involved in DA uptake in cortical and striatal neonatal rat astrocytes was examined by determining their IC_50_ values for the inhibition of transporter-mediated [^3^H]-DA uptake.

[^3^H]-DA uptake was more prominently inhibited in striatal than in cortical astrocytes by the selective DAT inhibitor, GBR12909, and NET inhibitor, desipramine, whereas inhibition by nortriptyline and amitriptyline was slight or nonsignificant in astrocytes from both brain regions. Corticosterone had no significant effect on [^3^H]-DA uptake in both striatal and cortical astrocytes, whereas the PMAT inhibitor, D22, inhibited [^3^H]-DA uptake similarly in both striatal and cortical astrocytes.

### 2.5. Changes in mRNA Expression of Plasma Membrane Monoamine Transporter and Norepinephrine Transporter in Cultured Neonatal Rat Astrocytes after 24 h Treatment with Apomorphine, Haloperidol, and L-DOPA

Cortical and striatal astrocyte cultures of neonatal rats were treated with apomorphine, haloperidol, and L-DOPA for 24 h. First, the cell viability after exposure to these three compounds was examined by MTS, Figure 5e, f. The astrocytes’ viability was not affected in comparison to the control even with the highest concentration used, 150 µM. We proceeded to perform qPCR analysis of astrocyte cell cultures after a 24 h treatment with the chosen concentration of 100 µM (Figure 5a–d).

Apomorphine induced the greatest fold change in mRNA expression in both striatal and cortical astrocytes of neonatal rats (Table 1). Apomorphine upregulated mRNA expression of NET mRNA in both cortical and striatal astrocytes but had an effect only on PMAT mRNA expression in cortical, not in striatal, astrocytes. Haloperidol and L-DOPA induced downregulation of PMAT, but the result was statistically nonsignificant (Table 1).

## 3. Discussion

In the present study, we examined the pharmacological and molecular characteristics of DA uptake in cortical and striatal astrocytes of neonatal rats to characterize the involvement of astrocytes from these two brain regions in the dopaminergic tripartite synapse. Additionally, we examined whether mRNA expression of the observed DA uptake transporters in astrocytes is sensitive to treatment with dopaminergic drugs and whether this is brain region-dependent.

DA uptake in cortical astrocytes of neonatal rats has been studied previously, but there is little research concerning the characterization of striatal DA uptake, particularly compared to the DA uptake in the cortical brain region. Research from Inazu et al. [37,38] and Pelton et al. [60] on cortical astrocytes of neonatal rats suggest that DA uptake is time, temperature-, concentration- and Na^+^-dependent as well as inhibited by ouabain, a Na^+^/K^+^-ATPase inhibitor; however, Inazu et al. have not found the DA uptake to be saturable at a 1 mM concentration of radiolabelled DA [38]. Similarly, Hansson et al. [61,62] and Hösli et al. [34] reported that astrocytes do not show saturable kinetics for DA, which they found to be Na^+^-independent, leading them to question the existence of a high-affinity carrier system in astrocytes. Our results suggest, similar to the results of Inazu et al. and Pelton et al., that both striatal and cortical astrocyte DA uptake is time-, temperature- and concentration-dependent. In the presence of nM concentrations of DA, uptake is significantly reduced by the absence of sodium ions, both in the uptake medium as well as by the inhibition of the Na^+^/K^+^-ATPase by 1 mM concentration of ouabain in both brain regions, indicating a presence of an active carrier system in astrocytes from both brain regions. The capacity of active carrier systems, such as DAT and NET, for DA transport is exceeded in the presence of higher concentrations of DA when the uptake 2 transporters, such as the OCTs and PMAT, become vital, due to their high-capacity, albeit low-affinity, for DA. Astrocytes from both brain regions display the ability to take up DA in the millimolar range, which indicates the prominent involvement of high-capacity transporters in both brain regions. Interestingly, striatal-specific DA uptake appears to be saturable, albeit at high concentrations of DA in the millimolar range, whereas cortical astrocytes-specific DA uptake appears to be nonsaturable. Based on these results, we presume astrocytes from both brain regions possess uptake 1 and uptake 2 carrier systems. Striatal astrocyte DA uptake is more dependent on uptake 1 transporters, whereas cortical astrocyte DA uptake is mediated primarily by facilitated diffusion. Nonetheless, astrocytes from both brain regions are capable of continuous DA uptake in the presence of higher concentrations (mM) of DA. As synaptic concentrations of DA have been observed to reach the millimolar range [63], the present study highlights the importance of astrocytes in the removal of excess DA from the synapse as well as DA homeostasis, serving as a sort of neuronal back up system.

To further identify which particular DA carrier system is involved in astrocyte DA uptake, we performed qPCR analysis. The presence of the main neuronal active DA uptake carrier system, DAT, in astrocytes has been reported by Asanuma et al. [39] in primary striatal astrocyte cultures of neonatal rats and in primary striatal cultures of neonatal BALB/c mice by Karakaya et al. [57]. The mRNA expression of DAT in astrocyte cultures from both brain regions in the present study appears to be low. Interestingly, striatal astrocytes exhibited greater mRNA expression of DAT compared to cortical astrocytes, although the expression of DAT appeared to be more prominent in cortical, rather than striatal neonatal rat tissue. GBR12909 inhibited striatal DA uptake but had less of an effect on cortical DA uptake. The correlation between levels of RNA and protein products of specific genes may vary [64]. As low mRNA levels of DAT may not reflect its protein expression and role in astrocyte DA uptake, strong inhibition of nanomolar concentrations of DA uptake in striatal astrocytes by DAT specific inhibitor, GBR12909, indicate its presence in our striatal astrocyte cultures. Nonetheless, our qPCR data show a more prominent mRNA expression of NET in astrocyte cultures from both brain regions, similar to the results of Takeda et al., who reported only NET, and not DAT, mRNA in cortical astrocytes of neonatal rats [24]. Its significance in both striatal and cortical astrocyte DA uptake is supported by the significant inhibition of DA uptake by desipramine, a NET selective inhibitor (pK_i_ 8.1–8.7 [25,65]). Interestingly, nortriptyline, a NET selective, but less potent, inhibitor (pK_i_ 8.2 [66,67]) had little effect on cortical compared to striatal DA uptake. The least potent NET inhibitor of the tricyclic antidepressants, amitriptyline, (pKi 6.5 [68]) produced only a slight inhibition of DA uptake in striatal, not cortical, astrocytes. Although we may presume astrocyte DA uptake is mediated to some extent by NET in both brain regions, its role may be of greater importance in striatal than in cortical astrocytes of neonatal rats.

Inhibition of the nonspecific uptake carriers, such as the OCTs and PMAT, was less potent. Specifically, the sensitivity of [^3^H]-DA uptake to the inhibition by corticosterone, an OCT inhibitor, was very low in astrocytes from both brain regions. Together with our qPCR data, this suggests a low involvement of the OCTs. PMAT is relatively insensitive to corticosterone (Ki = 450 µM [69]) but more potently inhibited by D22 than the OCTs [70] (pKi 7.0 [33,71]). D22-induced inhibition of DA uptake at higher concentrations (around 10 µM), indicates that PMAT could be involved in astrocyte DA uptake in both striatal and cortical astrocytes, albeit to a limited extent. Additionally, it is worth noting the effect of desipramine on [^3^H]-DA uptake may be attributed to reported nonspecific inhibitory effects of high concentrations of desipramine on other transporters, such as PMAT, which has been shown to be inhibited by a large array of antidepressants, albeit with affinities in the 5–200 µM range [71,72]. 

The overlapping expression and selectivity of monoamine neurotransmitter transporters as well as transporter inhibitors pose a difficult challenge in determining the contributions of each individual transporters in neurotransmitter uptake studies [73]. Nevertheless, results of the present study indicate neonatal rat striatal and cortical astrocytes are capable of DA uptake, mediated by similar uptake carrier systems, in the millimolar range, which primarily relies on NET and PMAT in cortical astrocytes, and on DAT, NET and PMAT in striatal astrocytes. 

Astrocytes participate in DA homeostasis and aid in DA removal from the synapse. Studies have shown astrocytes respond to various dopaminergic drugs [21]; however, research on whether astrocytes may serve as a potential therapeutic target of currently available treatment options is lacking. We used three different compounds: apomorphine, haloperidol, and L-DOPA for the treatment of astrocyte cell cultures in the present study. L-DOPA, a prodrug of DA administered to patients with PD, is metabolized to DA and supplements the low endogenous levels of DA. L-DOPA can interact with D1 or D2 receptors independent of its conversion to endogenous dopamine [74]. Haloperidol competitively blocks post-synaptic D2 in the brain, eliminating DA neurotransmission and leading to the relief of delusions and hallucinations that are commonly associated with psychosis [75]. Apomorphine is the oldest dopaminergic drug available for PD and remains the only drug with efficacy comparable to that of L-DOPA [76]. Like L-DOPA and DA, apomorphine acts as a potent, direct, and broad-spectrum DA agonist, activating all DA receptor subtypes, serotonin receptors, and α-adrenergic receptors [77]. Several studies have investigated the effects of apomorphine [56,78,79], haloperidol [50,52,54,80,81], and L-DOPA [39,82] on astrocytes. Apomorphine has been observed to enhance the biosynthesis of multiple trophic growth factors and is known for its ability to promote neuronal survival [56]. Haloperidol [54] and L-DOPA have been observed to induce a proinflammatory response. Particularly, L-DOPA has been investigated from the aspect of L-DOPA-induced dyskinesia associated with glial activation [83,84]. DA receptor susceptibility to dopaminergic drugs [75,85,86] as well as pathological conditions, such as hypoxia [87], have been explored by multiple studies. The role of dopaminergic drugs in astrocyte DA homeostasis and transport is, however, underexplored. Therefore, we have investigated whether they may affect the role of astrocytes in DA uptake, more specifically, the mRNA expression of transporters, NET and PMAT, involved in astrocyte DA uptake. In the present study, astrocyte mRNA expression of transporter, NET, was upregulated by treatment with apomorphine in astrocytes from both brain regions, whereas PMAT mRNA expression was upregulated only in cortical, but not in striatal astrocytes. Haloperidol and L-DOPA had no significant effect on mRNA expression of either transporter. Our findings indicate apomorphine may induce brain region-specific changes in astrocyte DA transport. 

In the present study, we confirmed that astrocytes cultivated from neonatal rat cortex and striatum differ in the molecular and pharmacological characteristics of DA transport. The transportation of DA into astrocytes from both brain regions depends on time, temperature, and exposure to sodium ions and ouabain; but, only DA uptake in striatal astrocytes is saturable and more sensitive to inhibition by desipramine and GBR12909. Astrocytes from both brain regions express DAT, NET, PMAT, and OCTs, but only apomorphine treatment affects their expression. Apomorphine increased the expression of NET and PMAT in cortical astrocytes but only NET in striatal astrocytes.

Although in vitro studies are a vital step in the study of new, unexplored concepts and areas, they inevitably pose some limitations. The prolonged lack of dopaminergic stimulation under nonphysiological conditions may affect the expression of transporters involved in DA uptake and may induce some changes in the characteristics of the observed astrocyte DA uptake. Nevertheless, the findings of the present study indicate that astrocytes from different brain regions have distinct characteristics in regard to DA homeostasis and are sensitive to treatment with dopaminergic drugs, such as apomorphine, which may open our horizons to potential new therapeutic targets in the treatment of neurodegenerative diseases, such as PD, that have yet to be managed successfully.

## 4. Materials and Methods

### 4.1. Materials

All tissue culture reagents, except fetal bovine serum, which was from Cambrex IEP GmbH (Wiesbaden, Germany), were obtained from Gibco Invitrogen (Paisley, Scotland, UK). [^3^H]-DA (2220 GBq/mmol) was purchased from Perkin Elmer (Waltham, MA, USA). The E.Z.N.A.^®^ HP Total RNA Kit was from Omega Bio-tek (Norcross, GA, USA), and the High Capacity cDNA Reverse Transcription Kit, TaqMan Gene Expression Assays, and TaqMan^®^ Gene Expression Master Mix were from Applied Biosystems (Carlsbad, CA, USA). GBR12909 was from Tocris (Bristol, UK), and decynium 22 (D22), corticosterone, L-DOPA, haloperidol, and apomorphine HCl were obtained from Sigma Aldrich (St. Louis, MO, USA). Nortriptyline HCl, desipramine HCl, and amitriptyline HCl were purchased from Sandoz (Cham, Switzerland). The CellTiter 96^®^ AQueous One Solution Cell Proliferation Assay (MTS) was obtained from Promega (Madison, WI, USA).

### 4.2. Animals and Primary Cell Culture Preparation

Astrocyte cell cultures were obtained from neonatal (3-day-old) rats, species *Rattus norvegicus*, strain Wistar, in accordance with the Administration of the Republic of Slovenia for Food Safety, Veterinary and Plant Protection issue U34401-23/2022/6. This study was approved by the National Veterinary Administration (approval numbers U34401-20/2017/2, approval date 20 June 2017 and U34401-23/2022/6, approval date 23 December 2022). Collectively, 30 animals were sacrificed using decapitation, and all necessary measures were taken to reduce discomfort and suffering of the animals according to the 3R principle. Astrocyte cultures were prepared as described [88,89]. Briefly, primary cultures derived from the striatum or cerebral cortex were grown in high-glucose Dulbecco’s Modified Eagle Medium (DMEM), containing 10% FBS, 1 mM pyruvate, 2 mM glutamine, and 25 μg/mL streptomycin at 37 °C in humidified 95% air/5% CO_2_. To reduce the number of contaminating microglial cells, confluent cultures were exposed to overnight shaking at 225 RPM. Medium containing detached cells was removed the next morning, and fresh growth medium was added. The whole procedure was repeated three times. Upon reaching confluence, the cells were seeded in 12-well plates and grown for an additional 3 weeks before being used for DA uptake or qPCR experiments. For cell viability experiments, cells were plated in 96-well plates and grown until reaching confluence (3–4 days).

### 4.3. Dopamine Uptake Experiments 

#### 4.3.1. Dependence of [^3^H]-Dopamine Uptake on Temperature, Time, and Concentration

Monolayer cultures in 12-well plates were preincubated for 30 min in the uptake buffer (25 mM HEPES, 125 mM NaCl, 4.8 mM KCl, 1.2 mM KH_2_PO_4_, 1.2 mM MgSO_4_, 1.4 mM CaCl_2_, and 5.6 mM glucose, pH 7.4) at 37 °C (total uptake) or at 4 °C (nonspecific uptake). To determine the time-dependence of DA uptake, cultured astrocytes were incubated with 30 nM [^3^H]-DA for different time intervals (0, 5, 10, 15, 20, 30, 45, 60 min). The concentration dependence of DA uptake was determined by exposing cultured astrocytes to different DA concentrations (up to 1 mM), as indicated in the Results, for 20 min. Specific DA uptake was calculated as the difference between the total and nonspecific [^3^H]-DA uptake. The apparent uptake velocity, V_max_, was determined using the Michaelis–Menten equation. Experiments were terminated by placing the plates on ice. [^3^H]-DA was removed quickly, and the plates were washed twice with ice-cold uptake buffer without Ca^2+^. The cells were subsequently lysed in 300 μL of 0.5 M NaOH. An aliquot (250 μL) of each sample was transferred to a scintillation vial to measure the radioactivity. The amount of transported DA was normalized to the total protein content, which was determined in the remaining aliquots (50 μL) of each sample using the Bradford method using the Bio-Rad Protein Assay (Hercules, CA, USA) measured with the Biotek Synergy HT Microplate Reader (Agilent, Santa Clara, CA, USA).

#### 4.3.2. Inhibition of [^3^H]-Dopamine Uptake

To determine the sensitivity of DA transport to different uptake inhibitors (corticosterone, D22, GBR12909, and antidepressants, nortriptyline, amitriptyline, and desipramine) or Na^+^/K^+^-ATPase inhibitor ouabain, cultured astrocytes were first preincubated in uptake buffer in the presence or absence of the inhibitor for 20 min at 37 °C. Astrocytes were then exposed to 30 nM [^3^H]-DA for 20 min. To assess the dependency of DA uptake on the presence of Na^+^, cultured astrocytes were incubated in the normal (Na^+^-containing) or Na^+^-free uptake buffer (125 mM NaCl in the uptake buffer was substituted with equimolar choline chloride ((CH_3_)_3_N(Cl)CH_2_CH_2_OH)) with 30 nM [^3^H]-DA for 20 min. Samples were harvested and processed for measurements of radioactivity and protein content as described above.

### 4.4. Quantitative Polymerase Chain Reaction (qPCR)

The total RNA from neonatal rat cortical and striatal tissue and cultured neonatal rat cortical and striatal astrocytes was extracted using EZNA HP Total RNA kit. Cell cultures were collected from 12-well plates, each well representing one sample, whereas striatal and cortical brain tissue samples were collected together from multiple animals. The striatal tissue was used whole. Due to its size, the cortical tissue was dissected into smaller sections, and the frontal part of the cortex was used for qPCR analysis. Brain tissue samples used for qPCR analysis were flash frozen in liquid nitrogen after excision and stored at −70 °C. Treated and nontreated astrocyte cell cultures plated onto 12-well plates were placed on ice and washed with sterile PBS buffer three times, which was then quickly and completely removed from each well. The plates were then stored at −70 °C until the total RNA extraction was performed. The quantity and quality of the extracted RNA was evaluated using absorbance as measured by the Biotek Synergy HT Microplate Reader on a Take3 microvolume plate. cDNA was synthesized using 1 µg of total RNA from each sample. qPCR analysis of cell culture samples from 12-well plates was performed in triplicates or quadruplicates, each well representing one sample. Experiments were repeated at least twice, and the analysis of tissue samples was performed using small sections (up to 30 mg) of previously flash frozen tissue from 3–5 animals. qPCR was performed using TaqMan Gene Expression Assays (PMAT (SLC29A4) Rn01453824_m1, NET (SLC6A2) Rn00580207_m1, DAT (SLC6A3) Rn00562224_m1, OCT1 (SLC22A1) (Rn00562250_m1), OCT2 (SLC22A2) (Rn00580893_m1), OCT3 (SLC22A3) Rn00570264_m1, and ß-actin (Rn00667869_m1)) according to the manufacturer’s instructions in the QuantStudio^TM^ 3 System (Thermo Fisher Scientific, Carlsbad, CA, USA). Expression of target genes was normalized to the expression of β-actin according to the equation [target/reference] = [EFF_reference_^Cq_reference_]/[EFF_target_^Cq_target_], where Cq is the quantification cycle and EFF is the amplification efficiency (expressed as a value between 1 and 2). EFF was determined with the LinRegPCR software (Version 2014.7) [90].

### 4.5. Cell Culture Treatment and Cell Viability

Cell viability was assessed using confluent astrocyte cell cultures plated in 96-well plates and treated with various concentrations of apomorphine, L-DOPA, and haloperidol (from 1–150 µM) for 24 h. The cell viability was measured using the CellTiter 96™ AQueous One Solution Cell Proliferation Assay (MTS) (Promega) according to the manufacturer’s instructions. For the qPCR analysis, confluent three-weeks-old neonatal rat cortical and striatal astrocyte cell cultures plated in 12-well plates were treated with 100 µM concentration of the three compounds, haloperidol, apomorphine, and L-DOPA, for 24 h, based on cell viability experiments.

### 4.6. Data Analysis

The uptake experiments and qPCR experiments on cell cultures were routinely carried out in triplicates or quadruplicates and each experiment was repeated at least twice. Triplicate and quadruplicate of each experiment consisted of the number of wells (3 or 4) used from the cell cultures plated onto 12-well plates. The total number of samples (n) represented the total number of wells used, pooled from multiple experiments. Similarly, in cell viability experiments, cell culture in each well plated onto 96-well plate represented one sample. All data are presented as arithmetic means ± SEM. GraphPad Prism 9.5 was used for processing, presentation, and statistical analysis of the data. The kinetic parameters (K_m_ and V_max_) and IC_50_ values were calculated using a nonlinear regression method using the GraphPad Prism software 9.5.0. The normality of the data distribution was investigated by the Shapiro–Wilk test. The comparison of data among groups was carried out using ANOVA (One-Way ANOVA or Brown–Forsythe ANOVA) with Dunnett’s or Dunnett’s T3 multiple comparisons post-hoc test. If only two groups of data were compared, the Mann–Whitney U test, an unpaired Student’s *t*-test or Welch’s *t*-test was used, depending on the normality of the data distribution and the variance. All *t*-tests were two-tailed. *p* values < 0.05 were considered statistically significant. 

## 5. Conclusions

In conclusion, our study investigated the molecular and pharmacological features of DA uptake in cortical and striatal astrocytes from neonatal rats, shedding light on their distinct roles in the dopaminergic tripartite synapse. Notably, we identified time-, temperature-, and concentration-dependent DA uptake mechanisms in both regions, with striatal astrocytes exhibiting saturable kinetics. Our mRNA expression analysis revealed low DAT levels but prominent NET and PMAT expression in astrocytes from both studied brain regions. Treatment with dopaminergic drugs influenced NET and PMAT expression differentially in cortical and striatal astrocytes.

## Figures and Tables

**Figure 1 ijms-25-00911-f001:**
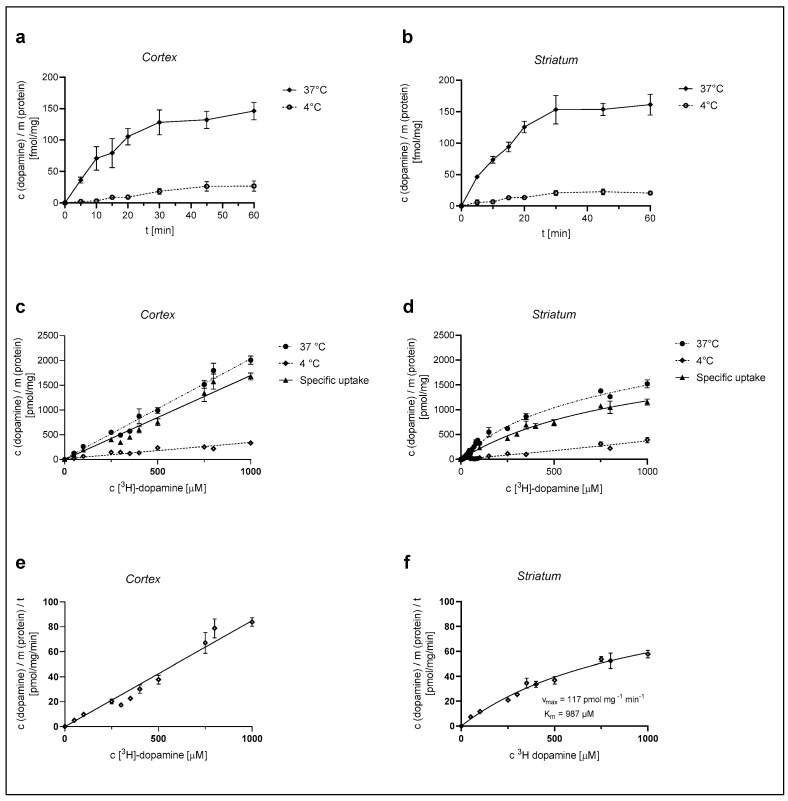
Dependence of [^3^H]-DA uptake on time, temperature, and concentration in cortical and striatal astrocytes of neonatal rats. Time dependence of total (37 °C) and nonspecific (4 °C) [^3^H]-DA uptake in cortical (**a**) and striatal (**b**) astrocytes, concentration dependence of [^3^H]-DA uptake (total, specific, and nonspecific) in cortical (**c**) and striatal (**d**) astrocytes and [^3^H]-DA uptake velocity in cortical (**e**) and striatal (**f**) astrocytes. Data are presented as mean ± SEM (*n* = 9) from three separate experiments.

**Figure 2 ijms-25-00911-f002:**
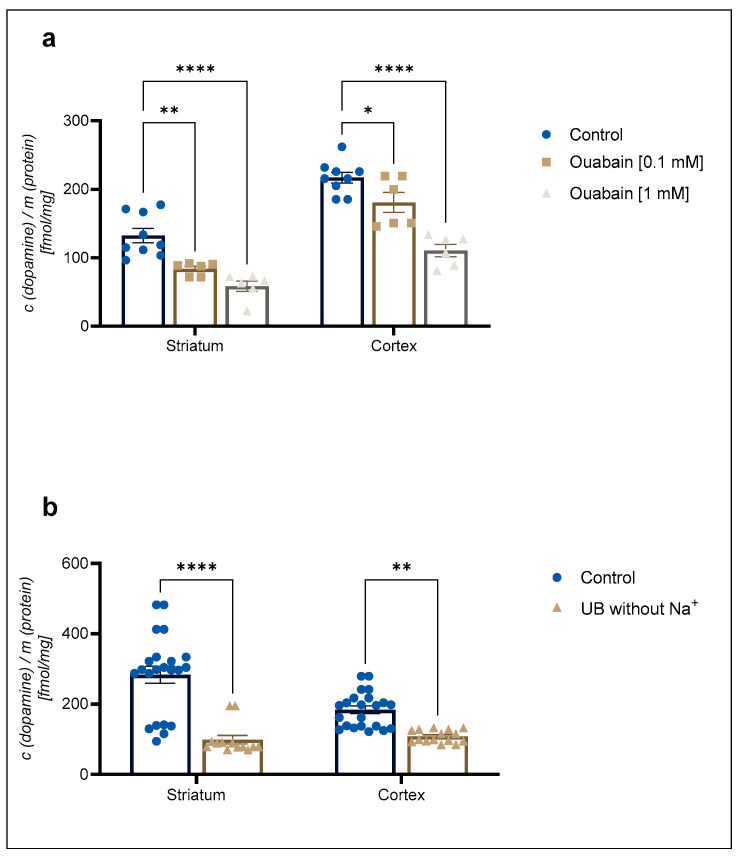
Inhibition of (**a**) 30 nM concentration of [^3^H]-DA uptake by 0.1 mM and 1 mM concentration of ouabain and (**b**) 30 nM concentration of [^3^H]-DA uptake in uptake buffer (UB) without Na^+^ in cortical and striatal astrocytes from neonatal rats. Results are presented as percent of the control; each bar expressed as mean ± SEM from three or more separate experiments ((**a**): *n* = 9, (**b**): *n* = 22). Statistical significance of reduction of DA uptake in comparison to the control was determined by One-Way ANOVA: * *p* < 0.05, ** *p* < 0.01, **** *p* < 0.0001. Difference between cortical and striatal DA uptake was nonsignificant (Student *t*-test, *p* > 0.05).

**Figure 3 ijms-25-00911-f003:**
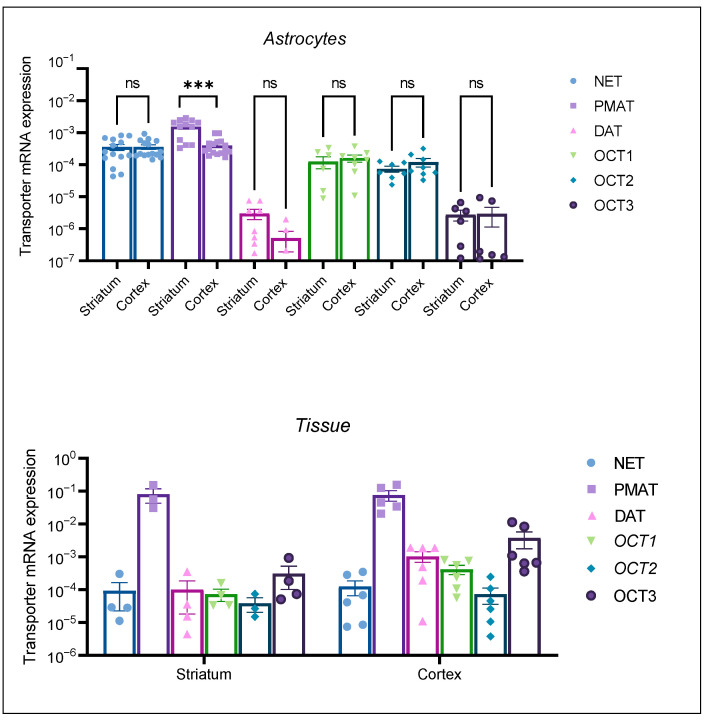
mRNA expression of transporters NET, PMAT, DAT, OCT1, OCT2, and OCT3 in striatal and cortical astrocytes cultured from neonatal rats (Astrocytes) as well as in striatal and cortical tissue samples of neonatal rats (Tissue). mRNA expression is normalized to the mRNA expression of the endogenous control, β-actin. Data are presented as mean ± SEM (astrocytes: *n* ≥ 6, from at least two separate experiments, tissue: *n* ≥ 4 from 4–6 animal tissue samples). Statistical analysis of comparative mRNA expression of transporters between striatal and cortical astrocytes (Astrocytes), was performed by multiple unpaired Mann–Whitney U tests with multiple testing correction two-stage step-up (Benjamin, Kriger, and Yekutieli) to control the false discovery rate (FDR) *** *p* = 0.001, ns—nonsignificant.

**Figure 4 ijms-25-00911-f004:**
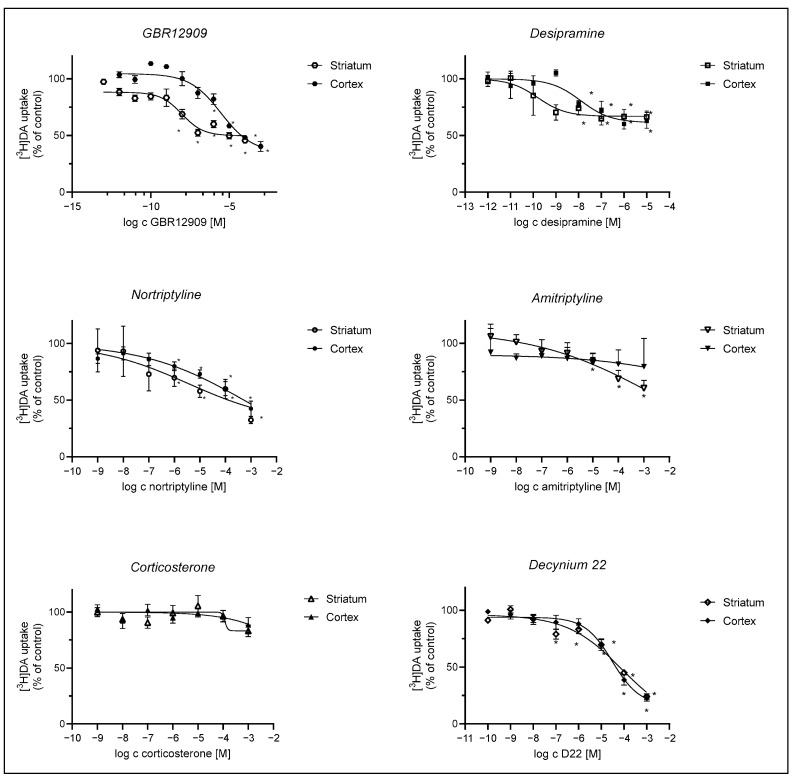
Inhibition of [^3^H]-DA uptake by GBR12909, desipramine, nortriptyline, amitriptyline, corticosterone, and D22. Astrocytes were preincubated with various compounds for 20 min, which was followed by a 20 min incubation with 30 nM [^3^H]-DA at 37 °C. Results are presented as percent (mean ± SEM) of the total [^3^H]-DA uptake of the control from three to four separate experiments carried out in triplicates (*n* ≥ 9). IC_50_ and pIC_50_ (calculated as the negative log of the corresponding IC_50_ values) of compounds inhibiting the total [^3^H]-DA uptake into cultured adult rat astrocytes were calculated from the corresponding inhibition curves. GBR12909: cortex: IC_50_ = 2.97 ± 2.46 µM, pIC_50_ = 5.5, striatum: IC_50_ = 0.00956 ± 0.00571 µM, pIC_50_ = 8.0, desipramine: cortex IC_50_ = 0.0125 ± 0.0137 µM, pIC_50_ = 7.9, striatum: IC_50_ = 0.000157 ± 0.000223 µM, pIC_50_ = 9.8; nortriptyline: cortex ns, striatum: IC_50_ = 3.5 ± 32.1 µM, pIC_50_ = 5.5, amitriptyline: ns, corticosterone: cortex and striatum: ns, D22: cortex—IC_50_ = 30 ± 17 µM, pIC_50_ = 4.5, striatum—IC_50_ = 81 ± 235 µM, pIC_50_ = 4.1. Statistical analysis of the difference in DA uptake in comparison to the control was carried out using an unpaired *t*-test with Welch control, * *p* < 0.05.

**Figure 5 ijms-25-00911-f005:**
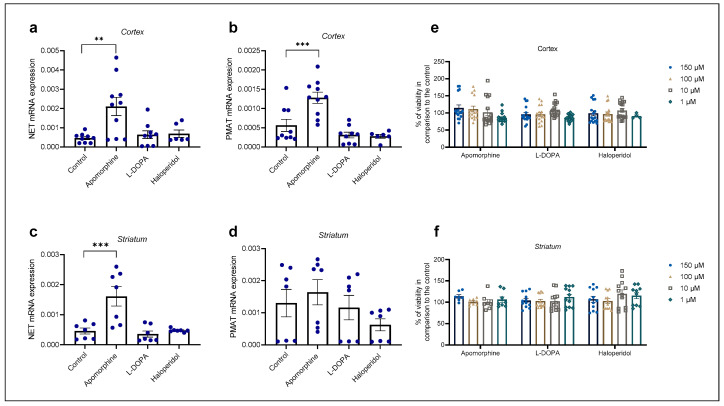
Changes in mRNA expression of transporters NET (**a**,**c**) and PMAT (**b**,**d**) after 24 h treatment with apomorphine, L-DOPA, and haloperidol. Data are presented relative to the expression of endogenous control, β-actin, as mean ± SEM of at least two separate experiments ((**a**,**b**): *n* = 9; (**c**,**d**): *n* = 7). Statistical analysis was performed using One-Way ANOVA with post-hoc Dunnett’s test (Cortex: NET: F(3,30) = 6.701, *p* = 0.0014, PMAT: F(3,30) = 13.88, *p* < 0.0001; striatum: NET: F(3,24) = 10.98, *p* < 0.0001, PMAT: F(3,24) = 1.392, *p* = 0.2695); ** *p* = 0.0013, *** *p* < 0.001. Cell viability of cortical (**e**) and striatal (**f**) astrocytes after 24 h treatment with various concentrations (1, 10, 100, 150 µM) of apomorphine, L-DOPA, and haloperidol presented as percent of the control, mean ± SEM ((**e**): *n* = 16, 4 separate experiments, (**f**): *n* = 12, 3 separate experiments). Differences in cell viability between treatments and control were determined as nonsignificant by One-Way ANOVA with post-hoc Dunnett’s test, *p* > 0.05.

**Table 1 ijms-25-00911-t001:** Fold changes in mRNA expression of transporters PMAT and NET after 24 h treatment with apomorphine, haloperidol, and L-DOPA. Significant (*p* < 0.05) fold changes in mRNA expression by apomorphine are emphasized by bold text.

Transporter	PMAT				NET			
Brain Region	Cortex		Striatum		Cortex		Striatum	
	Fold Change	*p* Value	Fold Change	*p* Value	Fold Change	*p* Value	Fold Change	*p* Value
Control	1 ± 0.4		1 ± 0.5		1 ± 0.2		1 ± 0.3	
Apomorphine	**2.3 ± 0.7**	**0.0008**	1.3 ± 0.5	0.8	**4.5 ± 1.3**	**0.02**	**3.5 ± 1.0**	**0.03**
Haloperidol	0.6 ± 0.2	0.7	0.9 ± 0.4	1.0	1.4 ± 0.5	0.8	0.8 ± 0.3	0.9
L-DOPA	0.5 ± 0.2	0.4	0.5 ± 0.2	0.4	1.5 ± 0.5	0.6	1.0 ± 0.2	1.0

## Data Availability

All data are contained within the manuscript. The data presented in this study are available on request from the corresponding author.

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
