# Peer review of "Cortical and Striatal Astrocytes of Neonatal Rats Display Distinct Molecular and Pharmacological Characteristics of Dopamine Uptake"

_ijms, 2024, doi:10.3390/ijms25020911_

Round 1

Reviewer 1 Report

Comments and Suggestions for Authors

The manuscript's results, in my opinion, are highly intriguing and helpful for the scientific community looking into the region-specific processes governing DA uptake in astrocytes. Make sure to indicate terminology clearly indicated in the Fig.3. Brain and renal tissue are the two different sorts. The striatum and cortex are the two regions of the brain that are selected. Since kidney tissue data are not further discussed in the manuscript's Introduction or Discussion sections, it is unclear why the authors decided to include them. Remove the material about kidneys, or include it in the Introduction and Discussion section the connection to brain tissue. The findings' conclusion might benefit from a little more specificity.

Comments on the Quality of English Language

Overall, English is of good quality.

Author Response

We have excluded the qPCR analysis data of the kidney sample as advised. Kidney sample was used as a positive control to validate our qPCR assays. The changes that were made in section 2.3. qPCR analysis of transporter mRNA expression in neonatal rat astrocytes from cortex and striatum of the Results are underlined:

2.3. qPCR analysis of transporter mRNA expression in neonatal rat astrocytes from cortex and striatum

We examined mRNA expression of high-affinity and low-affinity transporters involved in DA uptake in neonatal rat astrocytes from cortex and striatum as well as neonatal rat cerebral cortical and striatal tissue. qPCR was performed after extraction of total RNA from three weeks old confluent cultures and tissue samples for comparison of transporter mRNA expression between tissue (additionally used as a positive control) and astrocyte cell culture samples. 

Figure 3. mRNA expression of transporters NET, PMAT, DAT, OCT1, OCT2 and OCT3 in striatal and cortical astrocytes cultured from neonatal rats (Astrocytes) as well as in striatal and cortical tissue samples of neonatal rats (Tissue). mRNA expression is normalised to the mRNA expression of the endogenous control ß-actin. Data are presented as mean ± SEM (astrocytes: n ≥ 6, from at least two separate experiments, tissue: n ≥ 4 from 4-6 animal tissue samples). Statistical analysis of comparative mRNA expression of transporters between striatal and cortical astrocytes (Astrocytes), was performed by multiple unpaired Mann-Whitney t tests with multiple testing correction two-stage step-up (Benjamin, Kriger, and Yekutieli) to control the false discovery rate (FDR) *** p = 0.001, ns – non-significant.

mRNA expression profile of the transporters in neonatal rat tissue samples (Figure 3, Tissue) was in line with our expectations. OCT1 and OCT2 mRNA expression was lower than OCT3 mRNA expression in the two studied brain regions. PMAT tissue expression was prominent in both tissue samples. Active uptake transporters DAT and NET were expressed in both tissue samples. Conversely, astrocyte cultures from both brain regions displayed a distinct transporter mRNA expression profile, with OCT1 and OCT2 being more prominently expressed than OCT3. Among the studied transporters PMAT mRNA expression appeared to be the greatest in both brain tissue samples, as well in cortical and striatal astrocytes. Astrocyte striatal mRNA expression of PMAT was statistically significantly greater than cortical PMAT mRNA expression (p < 0.05) (Figure 3, Astrocytes). mRNA expression of uptake 1 transporter NET exceeded expression of DAT in astrocyte cell cultures from both brain regions, cortex and striatum. Interestingly, DAT mRNA expression was lower in striatal than in cortical neonatal rat tissue, whereas striatal astrocytes displayed low, however present mRNA expression of DAT, contrary to cortical astrocytes, where DAT mRNA expression was at the limit of detection of our qPCR method (Figure 3, Astrocytes). Differences in transporter expression in astrocytes from each brain region were determined separately by Brown-Forsythe ANOVA test with post-hoc Dunnett's T3 multiple comparisons test (Cortex: F=17.54 (5.00, 35.28), p < 0.0001, Striatum F=23.21 (5.00, 17.05), p < 0.0001). Striatal PMAT mRNA expression was significantly greater than all other transporters (p < 0.05), whereas NET mRNA expression was significantly greater than DAT, OCT2 and OCT3 (p < 0.05). Cortical astrocytes displayed no significant difference between mRNA expression of PMAT and NET, however mRNA expression of both transporters was significantly greater than expression of OCT1, OCT2, OCT3 and DAT. 

  1. We have made some additional concluding remarks in the discussion as advised:

In the present study we confirmed that astrocytes cultivated from neonatal rat cortex and striatum differ in molecular and pharmacological characteristics of DA transport.  Transport of DA into astrocytes from both brain regions depends on time, temperature, exposure to sodium ions and ouabain; but only DA uptake in striatal astrocytes is saturable and more sensitive to inhibition by desipramine and GBR12909. Astrocytes from both brain regions express DAT, NET, PMAT and OCTs, but only apomorphine treatment affects their expression. Apomorphine increased the expression of NET and PMAT in cortical but only NET in striatal astrocytes.

Although in vitro studies are a vital step in research of new unexplored concepts and areas, they inevitably pose some limitations. The prolonged lack of dopaminergic stimulation under non-physiological conditions, may affect the expression of transporters involved in DA uptake and may induce some changes in the characteristics of the observed astrocyte DA uptake. Nevertheless, the findings of the present study indicate, that astrocytes from different brain regions have distinct characteristics in regard to DA homeostasis and are sensitive to treatment with dopaminergic drugs, such as apomorphine, which may open our horizons to potential new therapeutic targets in the treatment of neurodegenerative diseases such as PD, that have yet to be managed successfully.

Reviewer 2 Report

Comments and Suggestions for Authors

In this study, the authors analyzed the molecular and pharmacological characteristic of cortical and striatal astrocytes from cell cultures obtained from neonatal Wistar rats.

Overall, the manuscript is well-written, and the research is presented clearly. In order to increase the quality of their work, the authors should address the comments presented below.

1) line 29: the authors state that astrocytes are “the most abundant cells in the CNS” – in general, the notion that glial cell vastly outnumber neurons (at least in the human brain) is being increasingly challenged (see this paper for reference: https://www.ncbi.nlm.nih.gov/pmc/articles/PMC5063692/); it would be better to state that astrocytes are “one of the most abundant cells in the CNS” in order to avoid confusion

2) in the materials and methods section, the authors need to describe, in detail, exactly how the brain tissue, from which primary cultures were derived, was sampled – how was the striatum delineated in neonatal rats; was the entire cortex taken or only certain parts of the cortex (if so, which parts?);

3) the authors need to clearly state how many rats were used in the experiment

4) data analysis – the authors may want to consider whether SEM is the appropriate way of presenting data variability in all experiments (could SD be better for any of the datasets?)

5) in the presentation of the results (especially in graph descriptions), it should be made more clear what exactly the sample size is referring to (e.g. Figure 1 states n = 9 – does this mean that 9 cell cultures were analyzed from 9 different rat brains, or does this refer to the total number of experiments, or something else entirely; this should be made clear to the reader)

6) lines 269 – 275: the authors should expand this section in order to better clarify the interpretation of their results; as it currently reads, the authors state that striatal and cortical specific DA uptake are saturable and non-saturable respectively, and from that the authors seem to conclude (“based on these results”) that both regions possess both uptake carrier systems. This conclusion does not seem to logically follow from the first statement. The authors should elaborate in the text how they reached this conclusion.

7) in the discussion, the authors may want to comment on/compare their finding to the findings of a very recent article by Nikolić B. et al., titled “Lasting mesothalamic dopamine imbalance and altered exploratory behavior in rat after mild neonatal hypoxic event” https://www.frontiersin.org/articles/10.3389/fnint.2023.1304338/abstract

Comments on the Quality of English Language

The overall quality of English is high. The text would benefit from minor corection in the use of the definite article. Some of the longer sentences would be made more comprehensible by using a comma (e.g. "Although they have traditionally been viewed as simple homogenous cells providing support to neurons we now recognize that astrocytes from different brain regions are heterogeneous"). 

Author Response

  1. We modified the text in the Introduction in line with the reviewer 1’s suggestion: Astrocytes, named after their star-like shape are one the most abundant cell types in the central nervous system (CNS) [1].

[1] von Bartheld, C.S.; Bahney, J.; Herculano-Houzel, S. The search for true numbers of neurons and glial cells in the human brain: A review of 150 years of cell counting. J Comp Neurol 2016, 524, 3865-3895, doi:10.1002/cne.24040.

  1. We explained in greater detail which brain regions of the neonatal rats were used in Materials and Methods (4.4.) as advised.

4.4. Quantitative polymerase chain reaction (qPCR)

The total RNA from neonatal rat cortical and striatal tissue and cultured neonatal rat cortical and striatal astrocytes was extracted using EZNA HP Total RNA kit. Cell cultures were collected from 12-well plates, each well representing one sample, whereas striatal and cortical brain tissue samples were collected together, from multiple animals. Striatal tissue was used whole, whereas due to its size cortical tissue was dissected into smaller sections and the frontal part of the cortex was used for qPCR analysis.

  1. We have included the number of animals used in the study as advised: Collectively, 30 animals were sacrificed by decapitation and all necessary measures were taken to reduce discomfort and suffering of the animals according to the 3R principle.
  2. The standard error of the mean (SEM) quantifies the precision of the mean. It is a measure of how far the sample mean is likely to be from the true population mean, whereas the standard deviation (SD) quantifies variability or scatter. In experimental studies using lesser number of samples SEM is usually used. In the present study we used the SEM instead of SD in Figures in order to make the graphic presentations easier to perceive.  For statistical analysis we used standard inferential statistics tools like Student t-test and ANOVA test which compare the means of the samples. In the kinetic uptake experiments SEM was used accordingly to previous research on uptake experiments [1-5]

References:

  1. Perdan-Pirkmajer, K.; Mavri, J.; Krzan, M. Histamine (re)uptake by astrocytes: an experimental and computational study. J Mol Model 2010, 16, 1151-1158, doi:10.1007/s00894-009-0624-9.
  2. Perdan-Pirkmajer, K.; Pirkmajer, S.; Černe, K.; Kržan, M. Molecular and kinetic characterization of histamine transport into adult rat cultured astrocytes. Neurochem Int 2012, 61, 415-422, doi:10.1016/j.neuint.2012.05.002.
  3. Perdan-Pirkmajer, K.; Pirkmajer, S.; Raztresen, A.; Krzan, M. Regional characteristics of histamine uptake into neonatal rat astrocytes. Neurochem Res 2013, 38, 1348-1359, doi:10.1007/s11064-013-1028-x.
  4. Inazu, M.; Kubota, N.; Takeda, H.; Zhang, J.; Kiuchi, Y.; Oguchi, K.; Matsumiya, T. Pharmacological characterization of dopamine transport in cultured rat astrocytes. Life Sci 1999, 64, 2239-2245, doi:10.1016/s0024-3205(99)00175-7.
  5. Krzan, M.; Schwartz, J.P. Histamine transport in neonatal and adult astrocytes. Inflamm Res 2006, 55 Suppl 1, S36-37, doi:10.1007/s00011-005-0031-3.

  1. To clarify the number of samples used in each experiment performed on astrocyte cell cultures we expanded the section 4.6 in Materials and Methods, as highlighted below by underlined text. Cell cultures were grown in 12-well plates and each well represented a separate sample of astrocyte cell culture. In each uptake or qPCR experiment cell cultures from three or four wells (n = 3 or n = 4) were exposed to the same conditions to represent the triplicate or quadruplicate, meaning a 12-well plate would consist of four triplicates or three quadruplicates. As the experiment was repeated at least twice the pooled number of the samples presented on the graphs is at least n ≥ 6.

4.6. Data analysis

The uptake experiments and qPCR experiments on cell cultures were routinely carried out in triplicates or quadruplicates and each experiment was repeated at least twice. Triplicate and quadruplicate of each experiment consisted of the number of wells (3 or 4) used from the cell cultures plated onto 12-well plates. The total number of samples (n) represented the total number of wells used, pooled from multiple experiments. Similarly, in cell viability experiments cell culture in each well plated onto 96-well plate, represented one sample.

  1. We expanded the discussion as advised. The changes that were made in the text are underlined:

The capacity of active carrier systems, such as DAT and NET, for DA transport is exceeded in the presence of higher concentrations of DA, when the uptake 2 transporters, such as the OCTs and PMAT become vital, due to their high-capacity, albeit low-affinity for DA. Astrocytes from both brain regions display ability to take up DA in the millimolar range, which indicates prominent involvement of high-capacity transporters in both brain regions. Interestingly, striatal specific DA uptake appears to be saturable, albeit at high concentrations of DA in the millimolar range, whereas cortical astrocytes specific DA uptake appears to be non-saturable. Based on these results, we presume astrocytes from both brain regions possess uptake 1 and uptake 2 carrier systems.

  1. We are thankful for the suggestion of the article Lasting mesothalamic dopamine imbalance and altered exploratory behaviour in rat after mild neonatal hypoxic event by Nikolić B. et al. We have included the reference in the discussion: DA receptor susceptibility to dopaminergic drugs [77,87,88] as well as pathological conditions such as hypoxia [89] have been explored by multiple studies. The role of dopaminergic drugs in astrocyte DA homeostasis and transport is however underexplored, therefore we have investigated whether they may affect astrocyte role in DA uptake, more specifically, the mRNA expression of transporters NET and PMAT, involved in astrocyte DA uptake.

[89] Nikolić, B.; Trnski Levak, S.; Kosic, K.; Drlje, M.; Banovac, I.; Hranilovic, D.; Jovanov Milosevic, N. Lasting mesothalamic dopamine imbalance and altered exploratory behavior in rat after mild neonatal hypoxic event. Frontiers in Integrative Neuroscience 2023, 17, doi:doi: 10.3389/fnint.2023.1304338.
